

# Quadriceps femoris cross-sectional area and specific leg strength: relationship between different muscles and squat variations

Filip Kojic[1], Saša Đurić[2], Igor Ranisavljev[3], Stanimir Stojiljkovic[3] and Vladimir Ilic[3]

[1] Teachers Education Faculty, University of Belgrade, Belgrade, Serbia
[2] Liberal Arts Department, American University of the Middle East, Kuwait City, Kuwait
[3] Faculty of Sport and Physical Education, University of Belgrade, Belgrade, Serbia

## ABSTRACT

**Background**. The aim was to determine the relationship between the cross-sectional area of the quadriceps femoris and strength performance in the deep and parallel barbell squat.

**Methods**. The sample included 16 university students (seven female, $24.1 \pm 1.7$ years). Muscle strength was expressed as external load, including the one-repetition maximum and the body mass segments involved (calculated according to Dempster's method). The cross-sectional area of the quadriceps femoris muscles was determined using ultrasound, while leg muscle mass was measured using the Bioelectrical Impedance method.

**Results**. The cross-sectional areas of the three vastii muscles and leg muscle mass showed moderate to strong correlation with external load in both squat types ($r = 0.509$–$0.873$). However, partial correlation (cross-sectional area of quadriceps femoris muscles were controlled) showed significant association only between leg muscle mass and deep squat ($r = 0.64$, $p < 0.05$). The cross-sectional area of the vastus lateralis showed a slightly higher correlation with external load in the parallel than in the deep squat ($r = 0.67$, $p < 0.01$ *vs.* $r = 0.59$, $p < 0.05$). The regression analysis extracted the vastus medialis cross-sectional area as the most important factor in manifesting strength (parallel squat: $R^2 = 0.569$; deep squat: $R^2 = 0.499$, both $p < 0.01$). The obtained results suggest that parallel squat strength depends mainly on the cross-sectional area of the vastii muscles, while it seems that the performance in the deep squat requires an additional engagement of the hip and back extensor muscle groups.

Corresponding author
Saša Đurić, sasa.duric@aum.edu.kw

# INTRODUCTION

The squat is one of the most popular and important exercises for developing strength and power and is often integrated into strength and conditioning training (*Toutoungi et al., 2000*; *Escamilla, 2001*). The exercise is performed in a closed kinetic chain involving the

hip, knee, and ankle joints and requires a significant level of hip and ankle mobility as well as stability of the lumbar spine (*Kritz, Cronin & Hume, 2009*). It is generally believed that the quadriceps femoris (QF) muscles, assuming four pennate muscles (rectus femoris-RF, vastus intermedius-VI, vastus medialis-VM and vastus lateralis-VL), make the greatest contribution to the squat motion, while the hip extensors, hip abductors, and ankle plantar flexors also play important roles (*Schoenfeld, 2010*).

In resistance training, there are numerous variations of the squat, such as the foot position, the position of the dumbbell (load), or the range of motion (*Schoenfeld, 2010*; *Clark, Lambert & Hunter, 2012*). The range of motion (ROM) during the squat, known as the knee flexion angle, is frequently modified in practice, with three types of squats most commonly used: (1) partial squat with ROM of 0–45 degrees (°), (2) parallel squat with knee flexion around 90°, and (3) deep squat with ROM over 100° (*Clark, Lambert & Hunter, 2012*). It has been shown that different ROM squat variations produce different dynamic and kinematic changes (*Drinkwater, Moore & Bird, 2012*), possibly indicating that engagement of upper-leg muscles could be dependent on movement amplitude prescription. The studies on the variability of muscle activation during different ROM squat variations mostly utilized electromyographic (EMG) analyses, and these reports are mostly inconclusive. While some findings indicate that a change in the amplitude of flexion does not cause changes in QF muscle activity (*Contreras et al., 2016*; *Da Silva et al., 2017*), others indicate a change in EMG activity of different QF muscle groups as a function of squat ROM (*Caterisano et al., 2002*; *Marchetti et al., 2016*). For example, *Contreras et al. (2016)* reported similar EMG responses of VL and the hip extensor muscles (gluteus maximus and biceps femoris) during both parallel and deep squats in resistance-trained females. However, in resistance-trained males, shorter variants of the squat (partial and parallel) elicited greater involvement of the VL, VM and RF muscles, while conversely the EMG amplitude of the gluteus maximus was greater during deep squats (*Caterisano et al., 2002*; *Marchetti et al., 2016*). The shortcoming of the aforementioned studies is that the contribution of the VI muscle is neglected because estimation of the EMG signals of the deep muscles, such as VI, cannot be performed directly with surface electrodes. The exception is the distal portion of the VI muscle, which is the only area available for surface EMG but can only be used for isometric contraction at low force levels (*Watanabe & Akima, 2011*). In addition, the majority of EMG studies have measured total muscle electrical activity and have not determined the partial contribution of the muscular and neural components. Considering that different squat modalities could accentuate intramuscular hypertrophy of the quadriceps (*Earp et al., 2015*; *Kubo, Ikebukuro & Yata, 2019*), which in turn could lead to different performance adaptations (*Mangine et al., 2014a*; *Mangine et al., 2014b*; *Wilhelm et al., 2014*; *Methenitis et al., 2016*) and alter potential knee injury risks (*Zebis et al., 2009*; *Toumi et al., 2013*; *Mangine et al., 2014a*), it is extremely important to evaluate and confirm the relationship between the development of different quadriceps muscle parts and squat strength in two fundamental ROM variations (*i.e.*, parallel *vs.* deep squat).

Although there are a considerable number of studies in the current literature that investigated the relationship between lower-body muscle size and squat strength (*Häkkinen et al., 1998*; *Secomb et al., 2015*; *Seitz et al., 2016*), most of them measured the outer thigh

**Table 1  Sample characteristics.**

| Variables | Mean ± SD |
|---|---|
| Age (years) | 24.1 ± 1.7 |
| BH (m) | 1.75 ± 0.08 |
| BM (kg) | 70.4 ± 12.03 |
| BMI (kg/m$^2$) | 22.7 ± 2.3 |
| SMM (kg) | 33.23 ± 7.60 |
| PBF (%)) | 16.48 ± 7.30 |
| LSMM (kg) | 9.3 ± 1.32 |

Notes.

BH, body height; BM, body mass; BMI, body mass index; SMM, skeletal muscle mass; PBF, percent body fat; LSMM, dominant leg skeletal muscle mass.

volume and did not provide data on the contribution of each quadriceps muscle parts in different ROMs squat performance. It should also be noted that previous research has assessed squat strength by one-repetition maximum test (1RM), which only reflects the weight lifted and does not take into account the mass of the other segments of the body (*i.e.,* head, trunk, and upper-leg). Hence, this could be a confounding factor when determining total QF muscle resistance, and incorporating both lifted weight and body segments would be more a reliable indicator of external load. Therefore, the aim of the study was to evaluate the relationship between the cross-sectional area of four quadriceps femoris muscles and external load measured during two major squat variations: deep squat and parallel squat. We hypothesized that different parts of the quadriceps muscle will have a different relationship with maximum strength in squats and that contribution of vastii muscles (VM, VL and VI) would be greater during parallel compared to deep squat.

## MATERIALS & METHODS

### Sample and study design

The sample included 16 university volunteers (nine males and seven females) with no experience of resistance training. During the sample selection, the study was advertised to all departments at the university (Table 1). Both sexes were included to reduce gender sampling disparity in sports science and medical research (*Costello, Bieuzen & Bleakley, 2014*).

Participants were not professional athletes and did not suffer from leg or back injuries. All participants voluntarily participated in the study and previously signed a written informed consent regarding the experimental procedure and potential risks. The study was approved by the Ethics Committee of the Faculty of Sport and Physical Education, University of Belgrade (protocol number: 2316/19-2) and conducted in accordance with the Declaration of Helsinki.

The strength tests were performed on two separate days, with the deep barbell squat on the first day and the parallel barbell squat on the second day. Before strength testing, two familiarization sessions were performed, where subjects received instructions on the proper squat technique for the deep and parallel squat exercises. During familiarization sessions, the subjects performed two controlled sets with eight to 10 reps of both squat
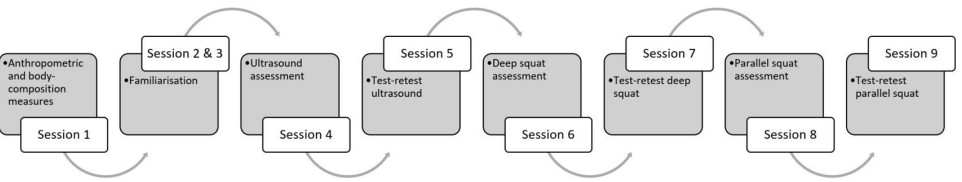

**Figure 1  Study protocol.**

variants with barbell on the shoulders, while being continuously monitored and corrected by experienced coach. Both squat variations were performed with the feet at shoulder width and with toes pointed forward or slightly outward. The barbell was placed in the high bar position across the shoulders on the trapezius, slightly above the acromion. To eliminate the negative impact of fatigue, the tests took place 48 h apart. Both tests were performed at approximately the same time of day (12–14 h). Between each test and retest session, the subjects had 48 h of rest, in order to avoid the effects of fatigue.

Anthropometry, body composition and quadriceps femoris (QF) cross-sectional measures were taken two days prior to first strength testing. Subjects were advised to avoid any form of physical activity 48 h prior to the testing, as well as not to consume alcoholic or caffeinated beverages 12 h before testing. The study protocol is shown in Fig. 1.

## Variables and protocols

Strength in both squat variations was assessed by one-repetition maximum test (1RM) using the standardized testing protocol (*Baechle & Earle, 2008*). The test was preceded by a 10-minute warm-up (light running and warm up exercises), followed by 8–10 repetitions of the exercise with a load of 50% 1RM and 2–3 repetitions of the exercise with a load of 60–80% 1RM. Each subject had five attempts to lift the maximum weight. Rest intervals between trials were set at 3 min. Deep squat was performed with maximum amplitude of flexion in the knee joint. The required knee angle in this position was 45 degrees (*Contreras et al., 2016*; *Kubo, Ikebukuro & Yata, 2019*), with the lumbar spine kept neutral. The height of the deep squat was measured for each participant individually. An elastic band was then stretched, which the participants had to touch during the squat to achieve the required angle of 45 degrees at the knee. Parallel squat was performed in a position where the femurs were parallel to the ground when the trochanter mayor and lateral epicondyle of the femur were at the same level (Fig. 2; *Earp et al., 2015*; *Contreras et al., 2016*; *Da Silva et al., 2017*). External load was calculated as the sum of 1RM and mass of body segments (head, trunk, arms and thighs) using the model proposed by *Dempster & Gaughran (1967)*. To determine the repeatability of the strength measurement, test-retest was conducted on ten participants on the two separate days.

Body height was measured with a Martin's portable anthropometer (Siber-Hegner, Switzerland) to an accuracy of 0.1 cm. Body composition variables, including skeletal muscle mass (SMM), percent body fat (PBF), and dominant leg muscle mass (LSMM), were measured with In-Body 720 (Biospace Co., Seoul, Korea) using Direct Segmental

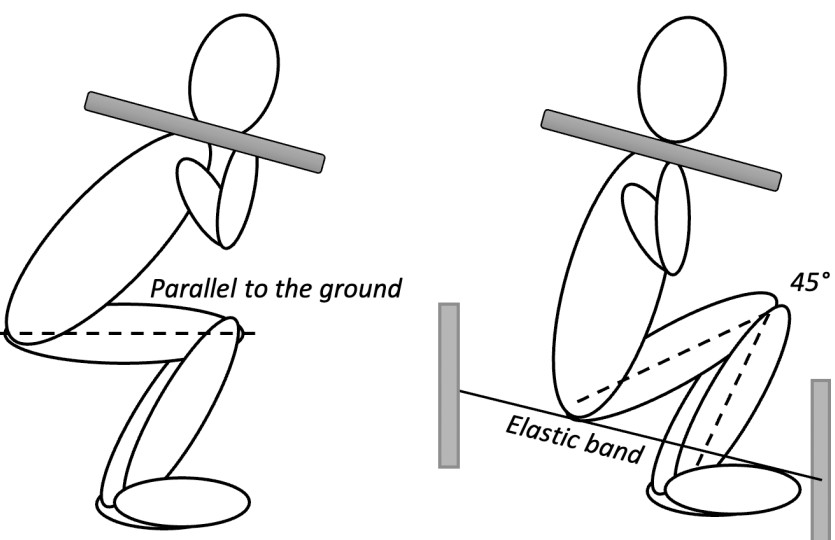

**Figure 2** Illustration of participants performing the deep squat and parallel squat.

Multi frequency–Bioelectrical Impedance Analysis (DSM–BIA method). Prior to testing, the subjects were instructed not to eat anything in the morning, avoid any kind of exercise 24 before body composition analyze and perform all physiological needs before the measurement. Subjects were in the standing position for at least 5 min prior to measurement for redistribution of body fluids. During the measurement all subjects were in light sport clothing and had no metal accessories. The subjects individually picked their dominant leg, based on the question "Which leg would you use to shoot the ball?" (*Van Melick et al., 2017*) and that variable was used for further analysis.

Image acquisition was performed by the experienced and previously trained radiology specialist. The cross-sectional area (CSA) was measured using an ultrasound scanner (Siemens Antares, Erlangen, Germany), with a linear probe of 7.5 MHz and expressed in square millimetres ($mm^2$). The 2D ellipse diagnostic method was applied. Imaging was conducted after about 10 min rest in supine position on right leg in full extension. The ultrasound probe was transversally placed on the skin with minimal compression with conductive gel used for acoustic coupling on the skins surface. The CSA of the rectus femoris (RF) was measured at the level of the three-fifths of the distance from the anterior superior iliac spine to the superior patellar border (*Seymour et al., 2009*), while the vastus intermedius (VI) CSA measurement was performed at one-half between spina iliaca anterior superior and the proximal border of the patella (*Ten Haaf et al., 2017*). The VL muscle CSA was measured at 50% of femur length, defined as the distance of 50% between the greater trochanter and the lateral condyle (*Franchi et al., 2018*). The vastus medialis (VM) was measured at the level of the distal portion above the medial side of patella. To determine the repeatability of the ultrasound measurement, test-retest was conducted on ten participants on the two separate days.

## Statistics

The Shapiro–Wilk and Levene'ş tests were used to assess the normality of the distribution and the homogeneity of variances, respectively. Reliability of the strength and ultrasound measures were accessed using the intra-class correlation coefficient (ICC). The independent samples $t$-test was used to determine the differences between male and female CSA of muscles and LSMM. To analyse the effects of sex and differences between external load of deep squat and parallel squat, the two-way ANOVA (model: Mixed between-within subjects) was used. Pearson's moment correlation was used to examine the association between external load, LSMM and CSA of 4 muscles forming QF (RF, VI, VM and VL). A partial correlation (controlling for QF muscles size) was performed to determine the influence of other lower-body muscle groups, except QF, on external load during both squat variations. The relationships were classified as trivial (<0.4), moderate (0.4–0.6), strong (0.6–0.8), very strong (0.8–1.0), and perfect correlation (1.0) (*Dancey & Reidy, 2007*). For discussion purposes, the correlation coefficients were directly compared with their 95% confidence intervals (*Vigotsky et al., 2019*). Additionally, to find the best predictive model of the quadriceps muscles for the external load of deep and parallel squat, a backward multiple regression was applied. Statistical analysis was performed using the IBM SPSS Statistics software package (Version 21, SPSS Inc, Chicago, IL, USA). All data are presented as means ± SD, where $p \leq 0.05$ was considered a statistically significant determinant.

## RESULTS

Excellent reliability was observed for both ultrasound (RF: ICC = 0.997, CI = 0.990–0.999, $p < 0.001$; VI: ICC = 0.995, CI = 0.979–0.998, $p < 0.001$; VM = 0.997, CI = 0.987–0.999, $p < 0.001$; VL: ICC = 0.996, CI = 0.982–0.999, $p < 0.001$) and strength (parallel squat: ICC = 0.997, CI = 0.987–0.999, $p < 0.001$; deep squat: ICC = 0.997, CI = 0.990–0.999, $p < 0.001$) measurements.

As expected, the CSA of QF muscle and LSMM was higher in males than females ($t = 3.449$, $df = 14$, $p = 0.004$, ES = 1.75 and $t = 5.824$, $df = 14$, $p = 0.000$, ES = 2.89, respectively). Males demonstrated significantly higher CSA of RF ($t = 2.238$, $df = 14$, $p = 0.042$, ES = 1.11), VI ($t = 3.626$, $df = 14$, $p = 0.003$, ES = 1.89) and VM ($t = 3.283$, $df = 14$, $p = 0.005$, ES = 1.71). A significant difference was not found only for VL ($t = 1.749$, $df = 14$, $p = 0.102$, ES = 0.92), although in females the average CSA of VL was lower for about 55 mm$^2$ (Table 2).

Male subjects achieved significantly higher external load for deep and parallel squat ($F = 24.882$, $df = 1$, $p = 0.000$, ES = 1.33). Subjects of both sexes demonstrated significantly higher external load for parallel compared to deep squat ($F = 44.136$, $df = 1$, $p = 0.000$, ES = 1.77). As the interaction between sex and squat variation was not statistically significant ($F = 0.036$, $df = 1$, $p = 0.852$, ES = 0.05), this proves that strength manifests in the same manner in both genders (Fig. 3).

Both deep squat ($p = 0.014$) and the parallel squat ($p = 0.004$) were significantly related to QF CSA, but the correlation was stronger for the parallel squat (see Table 3). VM and VL showed the strongest correlation with parallel squat ($p = 0.001$ amd $p = 0.005$,

**Table 2   Cross-section area of quadriceps muscles and leg muscle mass between males and females.**

| Muscles | Males Mean ± SD | Females Mean ± SD |
|---|---|---|
| Rectus femoris (mm²) | 383.67 ± 79.73[*] | 285.71 ± 95.49 |
| Vastus intermedius (mm²) | 333.44 ± 83.20[**] | 205,71 ± 46.59 |
| Vastus medialis (mm²) | 337.11 ± 67.57[**] | 242,29 ± 39.69 |
| Vastus lateralis (mm²) | 374.67 ± 72.45 | 321.00 ± 40.59 |
| Quadriceps femoris (mm²) | 1428.89 ± 223.04[**] | 1054.71 ± 204.48 |
| LSMM (kg) | 10.68 ± 1.02[**] | 7.52 ± 1.15 |

Notes.

[*] A significant gender difference ($p < 0.05$).

[**] A significant gender difference ($p < 0.01$).

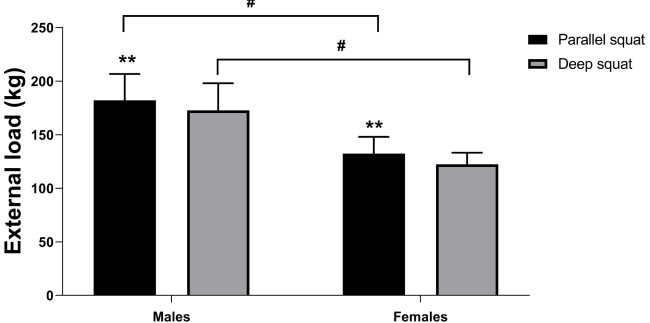

**Figure 3   Deep squat and parallel squat external load values for males and females.** Two asterisks (**)- indicate a significant squat variation difference ($p < 0.01$); a number sign (#) indicates a significant gender difference ($p < 0.01$).

**Table 3   Correlation between external load and cross-section area of quadriceps femoris muscles.**

| Muscles | Parallel squat r | Parallel squat p | Deep squat r | Deep squat p |
|---|---|---|---|---|
| Rectus femoris | .408 | .117 | .315 | .235 |
| Vastus intermedius | .546 | .029 | .509 | .044 |
| Vastus medialis | .754 | .001 | .706 | .002 |
| Vastus lateralis | .669 | .005 | .598 | .014 |
| Quadriceps femoris | .672 | .004 | .599 | .014 |

respectively), followed by VI ($p = 0.029$). For deep squat, the strongest correlation was observed with VM ($p = 0.002$), followed by VL ($p = 0.014$) and VI ($p = 0.044$). Only CSA of RF showed no significant association with external load measured during deep and parallel squat ($p = 0.267$ and $p = 0.104$, respectively; Table 3). LSMM was significantly correlated with both squat variation ($p = 0.000$). Even when a CSA of QF was controlled, LSMM was significantly correlated with deep ($r = 0.641$, $p = 0.026$), but not with parallel squat ($r = 0.530$, $p = 0.076$).

**A**

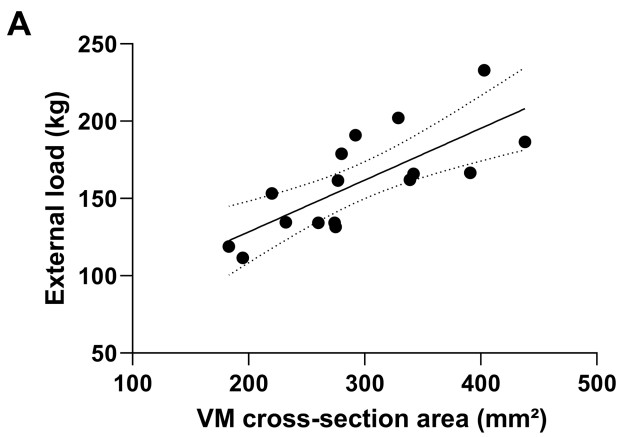

**B**

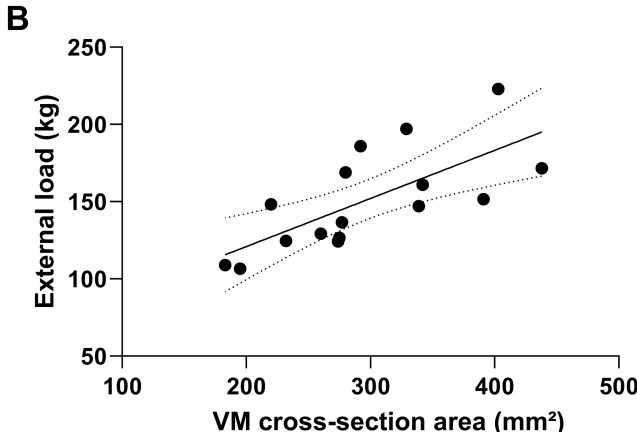

**Figure 4 External load prediction for parallel (A) and deep squat (B) based on the cross-section area of vastus medialis.** VM, vastus medialis.

Further regression analysis revealed that of the four predictors of quadriceps muscles, the best-fit model highlighted VM as the most important factor in the manifestation of strength in both squat variations. For external load of parallel squat, VM explained about 57% of variance ($R^2 = 0.569$, $p = 0.001$), with the equation for this model: $y = 0.3349x + 61.365$. On the other hand, the VM model explained about 50% of the external load deep squat ($R^2 = 0.499$, $p = 0.002$), with the equation: $y = 0.3114x + 58.609$ (Fig. 4).

## DISCUSSION

The study was conducted with the aim to determine contribution of quadriceps muscle parts in manifestation of strength during deep and parallel barbell squat. The present results indicate that there is a significant correlation between the CSA of QF and 1RM strength in both squat variations, but a stronger association was found with PS. Squat performance is largely dependent on the CSA of all three vastii muscles, with VM dimensions playing a

significant role in the manifestation of strength in both squat variations. The contribution of VL CSA appears to be more pronounced in the parallel compared to the deep squat.

We observed a significant association between lower-body muscle size and maximal strength in both squat variations. In addition, males demonstrated larger CSA of QF muscles and LSMM, as well as higher values of external load for both squat types. This implies that larger muscle sizes in males are the fundamental reason for significantly higher strength for deep and parallel squat. Pearson correlation coefficient revealed moderate to strong association between muscle size and squat strength ($r =$ 0.58–0.68). Similar results were obtained by *Suchomel & Stone (2017)* ($r =$ 0.64) and *Häkkinen et al. (1998)* ($r =$ 0.56–0.64) for males and females, indicating that QF muscle mass plays an important role in squat strength performance, however additional factors are also important (*i.e.,* other muscle groups, neural activity, muscle architecture). These statements are further confirmed by our results, as squat external load is more related to total leg muscle mass compared to the QF CSA. Moreover, QF CSA showed a rather stronger relationship with parallel than with deep squat ($r =$ 0.68, $p < 0.01$ *vs.* $r =$ 0.59, $p < 0.05$, respectively), while conversely, when CSA of QF was controlled, LSMM was significantly correlated only with deep squat. Considering that ROM significantly determines strength output in the squat (*Drinkwater, Moore & Bird, 2012*), we can assume that in addition to the QF, the other leg muscle groups are also significantly involved in the performance of squats with larger ROMs. Previous training studies with deep and half squats, showed that performing deep squats (ROM 0°–140°) influence higher relative increases in squat 1RM and significantly affect the volume of adductor and gluteus maximus muscles, in comparison to half squats (*Kubo, Ikebukuro & Yata, 2019*). However, the level of quadriceps hypertrophy was similar in both groups. On the other hand, *Bloomquist et al. (2013)* demonstrated greater QF hypertrophy after full ROM squats (0°–120°) compared to partial (0°–60°). These partial disagreements could be explained by different squats ROM prescription, which might affect the level of stimulus on different leg muscle groups.

In contrast to the vastii muscles, RF showed no significant association with 1RM in either squat variation. Apart from knee extension, RF acts as a hip flexor (*Robertson, Wilson & Pierre, 2008*), and it is quite possible that single-joint knee extension exercises are more suitable for training the RF (*Ema et al., 2016*; *Kubo, Ikebukuro & Yata, 2019*). The CSA of the VM showed the highest correlation with both deep and parallel squat, indicating a crucial role of VM muscle size in squat performance. From a practical point of view, this is of great importance considering that knee stability depends primarily on VM function and that VM asymmetry, as well as VM atrophy, lead to various knee problems including patelo-femoral pain syndrome (*Toumi et al., 2013*). The relationship between the CSA of VL and parallel squat was stronger compared to deep squat ($r =$ 0.65, $p < 0.01$ *vs.* $r =$ 0.55, $p < 0.05$), explaining why the CSA of total QF was more strongly correlated with the external load of parallel squat. In addition, both *Brechue & Abe (2002)* and *Secomb et al. (2015)* reported that the thickness of VL was a strong predictor of 1RM in back squat and squat jump performance, while *Earp et al. (2010)* indicated that leg power production and jumping performance were strongly dependent on the architecture of the gastrocnemius lateralis muscle rather than VL. *Nimphius, McGuigan & Newton (2012)*

showed a moderate relationship between VL muscle thickness and relative squat 1RM and emphasized that the performance of the muscle is strongly dependent on its architecture, including muscle fascicle length and pennation angle. The complexity of the leg strength and power performance phenomenon in multi-joint movements consisting of hip, knee, and ankle is even greater when the total EMG activity during squat variations is analysed. Using the muscle EMG approach in squat analysis, *Contreras et al. (2016)* and *Caterisano et al. (2002)* reported that the EMG values of VL and VM showed no significant differences during DS, PS and front squat. Moreover, EMG measurements support the importance of the hip extensors, especially the gluteus maximus in DS strength output, whose EMG amplitude showed a significant increase with squat depth (*Caterisano et al., 2002*).

### Strengths and limitations of the study

The main strength of this study is that this is the first study to evaluate the contribution of each quadriceps muscle part in different ROMs of squat performance. Most previous studies only measured outer thigh volume and did not provide this information. Moreover, we were able to evaluate the contribution of the VI, which was neglected in previous studies. Another strength of this study is that it provides more accurate information about muscle strength by calculating the exact external load, taking into account the mass of the other body segments in addition to the 1RM. This is very important when calculating muscle strength as it could be a confounding factor in determining total QF muscle strength.

A limitation of the present study investigating CSA and strength in deep and parallel squat includes the lack of data on the neural contraction component and muscle architecture. In addition, the CSA of all three vasti muscles (VI, VM and VL) was measured in the distal region. Previous studies have shown that CSA varies depending on the measurement position (*Noorkoiv, Nosaka & Blazevich, 2010*), so there is a possibility of partial variation in results, and we recommend that future studies include measurement of QF in the proximal, middle, and distal regions. The second limitation refers to the strength testing protocol, as the order of the tests was fixed, which could have resulted in increased fatigue from the first test session influencing strength in the second session. Therefore we strongly encourage that future studies apply a randomised strength testing design. Also, it should be noted that this is cross sectional research, based on correlation analysis, and longitudinal training studies are required to better explore how specific hypertrophy of QF regions is affected by squat depth variations. In addition, subjects require optimal hip and ankle mobility to perform deep squat safely and successfully, so this type of exercise is not suitable for everyone in practice (*Contreras et al., 2016*). Therefore, standardization of depth during deep squat is difficult to establish among the subjects, mainly due to varying levels of joint mobility and muscle flexibility. We instructed our subjects to go as deep as possible and reach 45° in the knees while keeping the lumbar spine neutral, which was quite difficult task for some subjects.

## CONCLUSIONS

In conclusion, the results prove that the external load in the parallel and deep squat is highly dependent on the CSA of the quadriceps femoris, while a slightly stronger association was

found in the parallel squat. The CSA of the VM has a fundamental role in manifesting strength in both types of squat and the highest correlation with both squat variations. VL showed a slightly higher correlation with parallel squat external load. The vastus intermedius showed an intermediate correlation with both variants of the squat, but slightly lower than VM and VL. We can assume that muscle strength during the parallel squat depends mainly on the CSA of three vastii muscles, while the deep squat performance requires additional function of other muscle groups, especially the hip and back extensors. From a practical point of view, the inclusion of both squat variations in resistance training program might be the best possible approach, while supplementary exercises should be implemented for optimal training of the rectus femoris due to the multi-joint function of this muscle.

### Funding
This article is a part of the project (No. III47015) funded by the Ministry of Education, Science and Technological Development of the Republic of Serbia. The funders had no role in study design, data collection and analysis, decision to publish, or preparation of the manuscript.

### Grant Disclosures
The following grant information was disclosed by the authors:
The Ministry of Education, Science and Technological Development of the Republic of Serbia: No. III47015.

### Competing Interests
The authors declare there are no competing interests.

### Author Contributions
- Filip Kojic conceived and designed the experiments, performed the experiments, analyzed the data, prepared figures and/or tables, authored or reviewed drafts of the paper, and approved the final draft.
- Saša Đurić analyzed the data, prepared figures and/or tables, authored or reviewed drafts of the paper, and approved the final draft.
- Igor Ranisavljev performed the experiments, prepared figures and/or tables, authored or reviewed drafts of the paper, and approved the final draft.
- Stanimir Stojiljkovic and Vladimir Ilic conceived and designed the experiments, authored or reviewed drafts of the paper, and approved the final draft.

### Human Ethics
The following information was supplied relating to ethical approvals (i.e., approving body and any reference numbers):
Ethics Committee of the Faculty of Sport and Physical Education, University of Belgrade approved this research (2316/19-2).

## Data Availability

The raw data is available in the Supplemental File.

## Supplemental Information

Supplemental information for this article can be found online at http://dx.doi.org/10.7717/peerj.12435#supplemental-information.

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
