# Peer review of "Quadriceps femoris cross-sectional area and specific leg strength: relationship between different muscles and squat variations"

_PeerJ, doi:10.7717/peerj.12435_

## Round 0.1 · original submission · Major Revisions

I appreciate your submission, but feel that some additional information is required before I send it out for external peer review. Please take on board these suggestions prior to resubmitting

1. include an abstract in the main document, as this appears to be missing

2. provide additional detail regarding your methods, in particular:
a. familiarisation of the participants with the squat exercises
b. inclusion of the picture to represent two squat positions and how a knee angle of 45° was achieved by all participants during the deep squat
c. standardisation of all relevant squat parameters such as bar position on the shoulders, stance width, as well as being more explicit was it a barbell squat or Smith machine squat?
d. reliability of the strength and ultrasound measures
e. rationale for the position at which the CSA of the different muscles was determined

---

## Round 0.2 · Major Revisions

While two reviewers and I see merit in your study and manuscript, there are also a number of areas that need improvement before this can be more seriously considered for publication and PeerJ.

·

Basic reporting

Throughout the manuscript, please change “gender” to “sex” when referring to biological sex differences.

Beginning of line 31 appears to be an extra space or underline
Line 79 compare should be “compared”
Line 151 accessed is misspelled
Line 171 measurement should be “measurements”
Line 205-206 “model with VM” should be changed to “the VM model”
Line 232 “in squat” should be “in the squat”
Liner 247 and earlier when you use the word “dimensions” it is unclear what you mean, please revise.

Experimental design

Line 83-84 “The sample included 16 randomly selected university students (9 males and 7 females)“ please provide more details about how they were randomly selected and the method of randomisation. Surely they volunteered versus being “selected”, per se. So, for example, was the study advertised to all departments at the entire university?
Line 94 “a two-familiarisation sessions” should just be “two familiarisation sessions” without a or a hyphen
Line 93-94 If I understand correctly, the order of the tests was fixed, and not randomised, if that’s the case please add as a limitation in the discussion that the fixed order of the strength testing could have introduced systematic error into the findings, such that fatigue from the first test session could have influenced strength in the second session.
Line 94-95 You mention there were two familiarisation sessions, but don’t describe them, please provide details of these sessions.
Line 103 Please clarify Figure 1. In the methods you state the participants were tested once on the deep squat and once on the parallel squat; however, Figure 1 shows a re-test of the two squats. Did all participants do both squat variations twice? If so, which test was used for correlational analyses with CSA? How far apart were retesting sessions?
Line 129 you state dominant leg skeletal muscle mass was measured, please describe the procedure for determining the dominant leg.

Validity of the findings

Line 191-196 you state a higher correlation was observed, but only report the p value but not the r values. Change the wording to indicate Table 3 shows the strength of the correlations and report p values for each finding. For example “Both deep squat (p = 014) and the parallel squat (p = 0.004) 1RM were significantly related to QF CSA, but the correlation was stronger for the parallel squat (see Table 3). Or similar.
Line 239-241 your statement that squatting to a 90 degree depth in the squat is optimal for quadriceps training is not supported but the references immediately preceding it. Bloomquist observed superior hypertrophy with deeper squats and Kubo observed no significant differences between groups. Please update your statement to reflect the data you cited or provide sufficient evidence that deeper squats are inferior for QF hypertrophy. Note that it is also a stretch to state that training studies are in line with your results as you performed a cross sectional, correlational analysis. Just because there were slight differences in 1RM to different depths with different QF muscles does not necessarily mean that those squats would result in differential hypertrophy in heads of the QF.

Limitations section: in addition to the non-randomised order of testing, please add to your limitations that this a cross sectional analysis, and that correlations should be further explored with longitudinal training studies to better explore how specific hypertrophy of QF regions is impacted by variations in squat depth.

Additional comments

Overall this is a solid study; however, it needs revisions to the methods primarily from the viewpoint of making sure that the reader would have sufficient information to replicate the study.

Secondly, some of the conclusions need to be made sufficiently conservative based on the nature of this analysis, and the extant research.

Finally, the limitations of this research need to be more clearly stated.

·

Basic reporting

This is a well-written manuscript with a very interesting methodological approach, population, and outcomes. The study determined the relationship between the cross-sectional area of the quadriceps femoris and strength performance in the deep and parallel barbell squat. The results indicate that there is a significant correlation between the CSA of QF and 1RM strength in both squat variations, but a stronger association was found with PS. Squat performance is largely dependent on the CSA of all three vastii muscles, with VM dimensions playing a significant role in the manifestation of strength in both squat variations. The contribution of VL CSA appears to be more pronounced in the parallel compared to the deep squat. I would like to praise the authors for the research topic and experimental design. All sections are well-described. I hope my comments will help the manuscript quality to be published in this Journal. I strongly recommend the authors to emphasize the novelties of the present study. Similar studies are found in literature. Moreover, the practical applications of the results are superficially explored. Moreover, it seems that the external load was evaluated also using Bioimpedance measures. This procedure is often poorly reproducible, I recommend to add reliability data in results section.

Experimental design

No comment.

Validity of the findings

No comment.

Additional comments

Page 6, Lines 21-22: “while the performance in the deep squat requires an additional engagement of the hip and back extensor muscle groups.” How could the authors confirm it based on the results?
Page 7, Line 53: “EMG estimates of the deep muscles”. Please check the gramma agreement here.
Page 9, Lines 83-84: The participants had no experience on strength training exercises and 1RM tests. I suppose this issue could be an issue (specially related do deep squat), although two familiarization sessions were performed. What do the authors think about that?
Page 10, Line 131: Different nutritional aspects as well as pre-test habits could play different roles in Bioelectrical Impedance. Could the authors better explain the pre-test recommendation as well as show the reliability of this measure?

---

## Round 0.3 · Minor Revisions

I thank the authors for their hard work in attending to virtually all of the reviewers comments. Please address the small number of minor comments of Reviewer 1 to further improve the quality of your manuscript.

·

Basic reporting

The authors effectively addressed all my comments in this section; however, there are a few minor English language errors that need to be corrected in the revisions, listed below.

"“Which leg would you use to shoot the ball?“ (van Melick et al., 2017) and that variable was used for further analyze (lines 140-142)."

The wording should be further "analysis" vs analyze.

"Previous training studies with deep and half squats, showed that performing deep squats (ROM 0º-140º) influence higher relative increase in squat 1RM and significantly affect the volume of adductor and gluteus maximus muscles, in compare to half squats (Kubo et al., 2019). However, the level of quadriceps hypertrophy was similar in both groups. On the other hand, Bloomquist et al. (2013) demonstrated greater QF hypertrophy after full ROM squats (0º - 120º) compared to partial (0º- 60º). These partial disagreements could be explained by different squats ROM prescription, which might affect the level of stimulus on different leg muscle groups (lines 245-249)."

In the first sentence, "increase" should be "increases" and "compare" should be "comparison".

"The second limitation refers to strength testing protocol, as the order of the tests was fixed, which could have resulted that increased fatigue from the first test session influenced strength in the second session. Therefore, we strongly encourage that future studies apply randomized strength testing design. Also, it should be noted that this is a cross sectional research, based on correlation analysis, and longitudinal training studies are required to better explore how specific hypertrophy of QF regions is affected by squat depth variations (lines 294-299)."

In the first sentence "the" should be placed before "strength testing", "that" should be changed to "in", and "influenced" should be changed to "influencing". In the second sentence "a" should be placed after "apply". In the third sentence "a" should be removed prior to "cross".

Experimental design

The authors effectively addressed all my comments in this section

Validity of the findings

The authors effectively addressed all my comments in this section

Additional comments

Thank you for revisions.

·

Basic reporting

The sections were improved after the first revision. I have no additional comments.

Experimental design

The sections were improved after the first revision. I have no additional comments.

Validity of the findings

The sections were improved after the first revision. I have no additional comments.

Additional comments

The sections were improved after the first revision. I have no additional comments.

---

## Round 0.4 · accepted · Accept

Thanks for your efforts in attending to the comments of the reviewers. I am happy to recommend this paper for publication in PeerJ.